# The Socio-Ecological Factors of Physical Activity Participation in Preschool-Aged Children with Disabilities

**DOI:** 10.3390/healthcare13091081

**Published:** 2025-05-07

**Authors:** Ming-Chih Sung, Mohammadreza Mahmoudkhani, Byungmo Ku

**Affiliations:** 1School of Kinesiology, Rongxiang Xu College of Health and Human Services, California State University, Los Angeles, CA 90032, USA; dsung@calstatela.edu; 2Laboratory of Physical Activity Behavior Change for Individuals with Disabilities, Yong-In University, Yongin-si 17092, Republic of Korea; mahmoudkhani@ut.ac.ir; 3Department of Special Physical Education, Yong-In University, Yongin-si 17092, Republic of Korea

**Keywords:** exercise, family, grandparents, health, physical therapy, siblings

## Abstract

Background: To effectively promote physical activity (PA) participation in preschool-aged children with disabilities (PACD), a comprehensive understanding of the associated factors is necessary. Consequently, this study aims to examine the factors influencing PA participation in PACD using the socio-ecological model. Methods: The Disability Status Survey 2020 in South Korea has been used for the current study. PACD aged below five years were selected, resulting in 5825 children. Variables were selected across each level of the socio-ecological model: (1) intrapersonal level (child’s sex, chronic condition, disability level, perceived health), (2) interpersonal level (siblings’ status, grandparents’ status, satisfaction with number of friends), (3) organizational level (enrollment in physical therapy, enrollment in occupational therapy), and (4) environmental level (residential area, government support). The decision tree analysis was conducted using chi-square automatic interaction detection to examine multi-level associated factors of PA participation in PACD. Results: The decision tree analysis produced a three-level model with six terminal nodes. In the study sample, 48.4% of participants reported participating in PA. The most common location for PA was at home, with balance exercises being the most frequent type of activity, followed by stretching. Frequent barriers to regular PA participation included a lack of professionals who are knowledgeable about disabilities and the absence of disability-specific programs. The decision tree analysis identified living with grandparents, the child’s sex, having a sibling, and enrollment in physical therapy as common predictors of PA participation in PACD. Conclusions: This study’s strength lies in its thorough examination of the socio-ecological factors influencing PA participation in PACD. The associated factors span both intrapersonal and interpersonal levels. To enhance PA participation among PACD, interventions should target these levels.

## 1. Introduction

Participation in physical activity (PA) is essential for overall well-being across all stages of life, including early childhood. Regular PA provides well-documented health benefits, such as enhanced cardiovascular and skeletal health, improved cognitive function, and greater social and psychological well-being [1,2]. Various campaigns, such as the Physical Activity Guidelines for Americans and recommendations from the World Health Organization (WHO), outline PA guidelines for young individuals, including those with and without disabilities [3,4]. A systematic review indicates that PA supports motor and cognitive development, as well as physical and psychological health in infants and toddlers [5]. More recently, another systematic review has shown that preschool-aged children with disabilities (PACD) can improve muscular fitness and motor skills through PA participation [6].

Although there is no international consensus on PA guidelines, several official recommendations encourage toddlers and preschoolers to engage in at least 180 min of PA per day [7]. A systematic review of 20 independent studies confirmed that toddlers’ PA levels vary widely, ranging from 72.9 to 636.5 min per day across studies [8]. While preschool-aged children without disabilities generally comply with PA guidelines [8], PACD tend to show physical inactivity [9]. Compared to preschool-aged children without disabilities, PACD participate in less PA [10] and physical play [11]. This difference in PA may stem from various factors, such as disability-related impaired physical function, a lack of resources and information about accessible facilities and programs, or negative perceptions and attitudes from the public [12,13]. Moreover, studies have indicated that, similar to their peers without disabilities, the levels of PA tend to decrease as children grow older [14].

Promoting PA participation in PACD requires examining the factors associated with it. A key approach to identifying these factors is applying the socio-ecological model, which offers a comprehensive framework for understanding the complex interplay between personal and environmental influences on health-related behaviors, including PA [15]. Only a few studies have specifically focused on individuals with disabilities [16,17]. A study indicated that proximity to parks and schools, along with active parental encouragement and support, significantly enhances PA levels among children with autism spectrum disorder [18]. A systematic review by Sutherland and colleagues identified 48 correlates of PA, framed within the socio-ecological model, in children and adolescents with intellectual disabilities [19]. The results showed that the identified correlates predominantly focused on intrapersonal-level factors, such as motor development, which was positively associated with PA.

While previous studies have provided valuable insights into how various factors within the socio-ecological model influence PA participation in the disability community, limited research has focused on these factors among PACD. Therefore, this study aimed to examine the associated factors of PA participation in PACD using the socio-ecological model.

## 2. Methods

The Disability Status Survey in South Korea has been implemented based on Article 31 of the “Welfare of Persons with Disabilities Act” and Article 18 of the “Enforcement Decree of the Welfare of Persons with Disabilities Act”. Surveys have been conducted from the first month in 1980 to the eleventh month in 2020. The purpose of the survey is to understand the population of people with disabilities and the incidence of disabilities, assess their living conditions and welfare needs, and produce foundational data for the establishment and implementation of both short-term and long-term welfare policies for people with disabilities. The survey is conducted every three years, with the 2020 survey being the most recent publicly available data source as of July 2024. The official report of this survey can be found on the following website: https://www.mohw.go.kr/board.es?mid=a10411010100&bid=0019&tag=&act=view&list_no=369030 (accessed on 8 July 2024). To gather responses from PACD, the survey employed a caregiver-proxy approach. Specifically, data were collected through home visits, during which trained investigators conducted structured interviews with caregivers of preschool-aged children with disabilities.

### 2.1. Data and Sample

Data from the Disability Status Survey 2020 were obtained from Ministry of Health and Welfare Korea Institute for Health and Social Affairs. The total number of the original data was 2,622,950. The age range of the participants was one to 100 years, and the median of the whole sample was 64.00 (SD: 18.32). For the purpose of the current study, participants aged below five years were selected, resulting in 5286 participants. There were no missing data in the variables used in the current study. Thus, all data were used in the analysis.

### 2.2. Measures

#### 2.2.1. Dependent Variable

In the current study, PA participation was defined as engagement in regular physical activity aimed at promoting health during daily life. The dependent variable, PA participation, was measured using the following question: ‘Have you regularly participated in physical activity for health management in the past year?’, with two response options of yes or no. Other PA-related information including frequency (week), duration (per opportunity), places, types, and barriers were also measured by the following questions: (1) How often do you exercise? (2) How many minutes do you exercise per session? (3) Where do you usually exercise? (4) What is the main type of exercise you participate in? (If multiple, based on the exercise you do most frequently), and (5) If you are not currently exercising, what is the main reason for this?

#### 2.2.2. Intrapersonal Level

At the intrapersonal level, several factors were assessed, including the child’s sex, chronic condition, disability level, perceived health, and parent’s income. The child’s sex was determined based on their parent’s report. The question posed was “What is your child’s sex?” with two response options of male and female. This contrasts with a national survey in the United States that includes multiple gender options, as the question about the child’s sex was limited to a binary choice. The status of chronic condition was determined based on the following question: “Do you have a chronic condition that has persisted for more than three months?”, with two response options of yes or no. The classification of disability level was based on the 2019 amendment of the “Disability Welfare Act” and its subordinate regulations for registered disabilities in South Korea. Previously classified into six levels (grades one to six), disability is now divided into two categories: “severe disability” (formerly grades one to three) and “less severe disability” (formerly grades four to six). Perceived health was assessed with the question, “How would you rate your usual state of health?” Participants could choose from five response options: (1) very good, (2) good, (3) normal, (4) bad, and (5) very bad. Responses 1 and 2 were grouped into a ‘good’ category, while responses 3 through 5 were categorized as ‘bad’. Monthly household income was categorized into two groups: less than or equal to KRW 300,000 and greater than KRW 301,000.

#### 2.2.3. Interpersonal Level

At the interpersonal level, status of brothers and sisters, status of grandparents, and satisfaction with number of friends were assessed. The status of brothers and sisters was measured using the question, ‘Please check all the household members residing in your home: Siblings’. It has two options: yes or no. Similarly, the status of grandparents was measured using the question, ‘Please check all the household members residing in your home: grandparents’. It has two options: yes or no. Satisfaction with number of friends was assessed with the question, “How are you satisfied with the number of friends you have?” Participants could choose from four response options: (1) very satisfied, (2) a little satisfied, (3) a little unsatisfied, and (4) very unsatisfied. Responses 1 and 2 were grouped into a ‘satisfied’ category, while responses 3 through 4 were categorized as ‘unsatisfied’.

#### 2.2.4. Organizational Level

Several factors were examined at the organizational level, including enrollment in physical therapy, enrollment in occupational therapy, and enrollment in preschool. The enrollment in physical therapy was measured using the question, ‘Please check if you receive regular physical therapy services’, with two response options: yes or no. The enrollment in occupational therapy was measured using the question, ‘Please check if you receive regular occupational therapy services’, with two response options: yes or no. The enrollment in preschool was determined by the question, “Please check if your child goes to pre-school”, with two response options: yes or no.

#### 2.2.5. Environmental Level

At the environmental level, the area of living was assessed through the question asking participants where they currently live, with three options provided: large city, medium-sized city, and town. The responses for large city and medium-sized city were grouped together under the category “urban”, while town was categorized as “rural”. Perceived government support was assessed through the survey question, “How much support do you feel you receive from the government or society after registering as a person with a disability?” Four response options were provided: (1) receiving a lot of support, (2) receiving some support, (3) receiving a little support, and (4) receiving no support at all. The responses were dichotomized into two groups: options 1 and 2, indicating ‘receiving support’, and options 3 and 4, indicating ‘receiving insufficient support’.

### 2.3. Data Analysis

Descriptive statistics, including means and percentages, were used to analyze the demographic information of participants. Chi-square tests were conducted on the demographic data to evaluate the homogeneity of the groups. These chi-square tests specifically focused on categorical variables within the demographic data to determine any associations between these variables and the two groups. To examine the associated factors of PA participation in PACD, decision tree analyses were conducted using chi-square automatic interaction detection (CHAID). The model was built using SPSS software, version 29.0, and to reduce the risk of overfitting [i.e., 10-fold], cross-validation was implemented. Cross-validation allowed for the evaluation of model performance during tree growth, helping to determine when further splitting no longer improved predictive accuracy. Node splits were selected based on chi-square tests, with a significance level of *p* < 0.05. The final model was selected by balancing predictive accuracy and model simplicity, consistent with best practices for decision tree analysis. The demographic characteristics of participants were included as covariates in the CHAID analysis. CHAID is a non-parametric statistical method used in data mining and decision tree analysis, designed to uncover relationships and interactions among categorical variables. The statistical model in this study was specified with the following criteria: (1) Pearson’s chi-square test; (2) adjustment of significance levels using the Bonferroni method; (3) a significance threshold of *p* < 0.05 for splitting nodes; (4) a minimum change in expected cell frequencies set at 0.001; (5) a maximum tree depth of three levels; (6) a minimum of 10% of the sample for parent nodes (*n* = 100) and 5% for child nodes (*n* = 50); and (7) at least 10 folds for cross-validation. Demographic information and health risk behaviors of the caregivers were utilized as input variables for the CHAID analysis.

## 3. Results

Among all participants, approximately 88.17% were four years old. Of these, 51.64% participated in PA. However, among the three-year-olds, 76.04% did not participate in PA (*n* = 625). Boys constituted 63.11% of the participants and were more active in PA compared to girls. According to Korea’s disability level criteria, 72.63% were categorized as severe. Children with mild disabilities were more active in PA compared to those in the severe group. Cerebral palsy was the most prevalent condition (35.28%), followed by language disorders (24.88%) and hearing impairments (12.22%). Among PACD, PA prevalence was highest in toddlers with autism spectrum disorder (62.23%), followed by those with cerebral palsy (47.29%) and language disorders (43.50%). Additionally, 42.94% reported that their household monthly income ranged from KRW 300 to 400. Table 1 presents detailed demographic information about the participants.

### 3.1. PA Participation

Notably, disability severity was classified into two categories: “severe” (*n* = 3839, 72.63%) and “mild” (*n* = 1446, 27.37%). As shown in Table 1, children with mild disabilities had a higher PA participation rate (63.83%) compared to those with severe disabilities (42.54%). Among all participants, 48.36% reported that they regularly engaged in PA over the past year. Of these, 43.6% participated in PA more than three times per week, with an average duration of 34.78 min per session. The most frequent location for PA was at home (38.99%), followed by disability-specific PA centers (18.54%), and public PA centers (15.58%). Furthermore, the most common type of PA was balance-related activities (32.55%), followed by stretching (25.94%) and bicycling (15.58%). Table 2 presents detailed information on PA participation among PACD. If participants reported that they did not currently participate in PA, the most frequent reason was a lack of professional knowledgeable about disabilities (28.78%), lack of disability programs (24.46%), and PA is not a priority (19.75%). Table 3 presents detailed information on barriers to PA participation in PACD.

### 3.2. Decision Tree Analysis

The decision tree analysis yielded a three-level model with six terminal nodes, identifying key determinants of physical activity (PA) participation among infants and toddlers with disabilities (PACD). In the study sample, 48.4% of participants reported participating in PA. The first split was based on whether the child lived with grandparents. Children living with grandparents had a significantly lower PA participation rate (8.0%, *n* = 47) compared to those not living with grandparents (53.4%, *n* = 2509). Among those not living with grandparents, the presence of siblings further differentiated PA participation. Children with siblings demonstrated a higher participation rate (71.5%, *n* = 1545) than those without siblings. Within the group with siblings, perceived health played an important role: children who perceived their health as “good” were more likely to participate in PA (62.6%, *n* = 938), whereas those with “bad” perceived health showed a much lower participation rate (2.5%, *n* = 26). Physical therapy attendance also emerged as a significant factor under the sibling group, with children receiving physical therapy being more likely to participate in PA than those who did not receive therapy. Additionally, after the second split, sex further differentiated participation, with girls showing higher PA participation rates compared to boys. To enhance clarity, a visual depiction of the decision tree structure is provided in Figure 1. Table 4 presents the percentage differences in socio-ecological factors related to PA between groups.

## 4. Discussion

Overall, multi-level factors such as living with grandparents, the status of siblings, the child’s sex, enrollment in physical therapy, and perceived health were significant factors influencing PA participation in PACD. Grandparents were the most strongly associated factor with PA participation among PACD. As grandparents can serve as another caregiver supporting young children’s PA, such as doing PA together or taking them to a park [20], it was expected that living with grandparents may be a positive influencing factor. However, PACD living with grandparents were less likely to participate in PA compared to their counterparts. Although further research is required to explore the mechanisms underlying this association, a few possible explanations can account for the low PA support in grandparents.

The first explanation may be differences in grandparents’ perspectives on the appropriateness of engaging in PA at home. A semi-structured interview conducted by Parrish and colleagues found that most families believed it was inappropriate for their preschool-aged children to engage in PA at home, largely due to restrictions imposed by grandparents against certain types of activities inside the house [21]. Barriers perceived by grandparents such as the lack of safe and accessible programs or facilities, and constraints related to transportation, financial resources, energy, and time may also contribute to lower PA participation among toddlers living with their grandparents [20]. A study revealed that co-residence with grandparents may lead to disagreement in supporting child’s PA, which in turn negatively affects grandparent’s PA support [22]. The negative influence of living with grandparents on PA in PACD aligns with previous research showing that children raised by caregiving grandparents are more likely to be overweight or obese compared to those raised by their parents [23]. A review study also found that grandparent involvement negatively affects children’s weight status [24]. Co-residence with grandparents has been positively associated with increased screen time in young children [25]. Physical challenges experienced by grandparents may further limit opportunities for engaging in physical activity with their grandchildren [26]. They perceived themselves as storytellers or historians rather than active playmates [27].

Among PACD who did not live with their grandparents, those with siblings were more likely to engage in PA than those without siblings. This finding aligns with previous studies showing that siblings influence PA levels in youth living in the same household [28]. Siblings may help shape and reinforce positive PA preferences, especially when multiple siblings are present, by creating opportunities for engagement and offering supervision during activities [29]. They serve as role models, and PACD may observe and imitate their siblings’ behaviors, including engaging in PA [30]. Interactions with siblings can positively influence the social support and physical development of children with disabilities [31]. Having siblings may foster a more relaxed and inclusive environment, encouraging greater physical engagement [32].

Another associated factor was enrollment in physical therapy. PACD who received regular physical therapy were more likely to participate in PA than those who did not. Regular participation in physical therapy may help children with disabilities integrate PA into their daily routines [33,34]. Studies have shown that consistent, structured interventions can lead to higher levels of PA beyond therapy sessions [35]. Physical therapy often includes training and educating parents and caregivers on how to support and encourage their child’s PA, which may create more opportunities for children to be active at home and participate in community activities [36]. For example, parents often serve as co-therapists by engaging in activities with their child at home, such as balance exercises or stretching.

The last important finding of the current study was the perceived barriers to PA for preschool-aged children with disabilities. This study revealed that the most significant barriers to PA participation PACD include a lack of professionals who are knowledgeable about disabilities, an absence of disability-specific programs, and the perception that PA is not a priority. These findings are consistent with the reports from [13], who indicated that individuals with disabilities face over 100 barriers that hinder their PA participation, as compared with their typically developing peers. These barriers consist of limited information about accessible programs, negative attitudes from the public, and lack of knowledge, education, and training among fitness professionals [13]. Further, a systematic review analyzed factors related to PA participation among individuals with disabilities based on the socio-ecological model, highlighting institutional barriers. such as program availability, as well as the knowledge of people within institutions/organizations encountered in this population [37]. In Asian culture particularly, PACD might spend more time receiving treatment and other therapies (e.g., speech therapy) [38], and might force parents to restrict the amount of time their child spent in PA participation [39]. These findings highlight the multifaceted barriers to PA among PACD and emphasize the need for interventions that directly address these challenges. Programs should be developed specifically to include preschool-aged children with various disabilities, with a focus on adaptations and modifications tailored to individual needs. Educating parents and caregivers about the critical role of PA in the physical, cognitive, and emotional development of children with disabilities is also essential.

Several limitations should be considered when interpreting the findings of this study. First, PA participation was based on caregiver-reported data, which may introduce reporting bias or inaccuracies due to subjective perceptions. However, the large sample size in the current study may help offset this limitation. Second, this study was conducted within a South Korean context, where the involvement of grandparents in caregiving is culturally prominent; therefore, the generalizability of the findings to other cultural settings may be limited. For example, grandparents living in the United States may place greater importance on PA and participate more actively in PA with their grandchildren. Third, PA participation was measured through self-reports rather than objective methods such as accelerometry, which may affect the precision of the results. Future studies incorporating objective PA measurements and exploring cross-cultural comparisons are recommended to build upon these findings. Another limitation is the lack of detailed information regarding the characteristics of physical and occupational therapy services. Depending on the type of therapy provided, the physical activity participation of PACD may vary. However, caregivers of PACD in South Korea often perceive physical therapy as an opportunity for their children to engage in physical activity. As the analysis of the 2020 data progressed, new data from 2023 were published; therefore, the current study may not reflect the most recent data.

## 5. Conclusions

The current study investigated PA participation among PACD. Among the 5286 participants, 51.06% did not engage in PA. PA participation was highest among toddlers with autism spectrum disorder (62.23%), followed by those with cerebral palsy (47.29%) and language disorders (43.50%). Among those who did not participate in PA, the most frequently reported barriers were a lack of professionals who are knowledgeable about disabilities (28.78%), a lack of disability-specific programs (24.46%), and the perception that PA is not a priority (19.75%). Male PACD living with grandparents were the least likely to participate in PA. Among PACD who did not live with their grandparents, those without siblings and those with poor perceived health were the least likely to participate. These findings suggest that PA participation in PACD is likely influenced by a range of multi-level factors. A major strength of this study is its comprehensive examination of both the barriers and facilitators to PA participation in this population. Notably, it is among the first studies to apply the socio-ecological model to identify factors associated with PA participation among PACD. The findings of this study have important implications for clinical and community practice. The positive association between physical therapy attendance and PA participation highlights the critical need to improve early access to physical therapy services for young children with disabilities. Expanding early intervention programs and ensuring that families can easily access therapy services may foster higher PA engagement from a young age. Additionally, given the negative association between living with grandparents and PA participation, there is a need for family-centered approaches that actively involve grandparents. Community providers and educators could develop inclusive programs that educate grandparents about the importance of PA, equip them with strategies to support active play, and provide accessible, family-oriented opportunities for PA. By addressing barriers at both the individual and family levels, clinicians, educators, and community organizations can more effectively promote PA participation in this vulnerable population.

## Figures and Tables

**Figure 1 healthcare-13-01081-f001:**
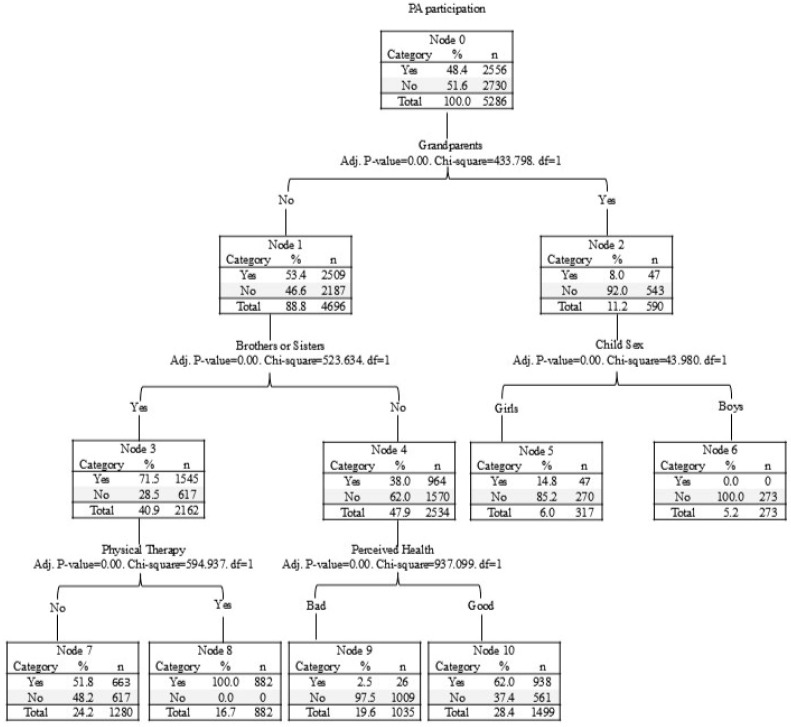
The results of decision tree analysis for socio-ecological factors of PA participation in PACD.

**Table 1 healthcare-13-01081-t001:** Demographic information of participants.

		All (*n* = 5286)	PAP (*n* = 2556)	No PAP (*n* = 2730)	χ^2^	*p*-Value
Child age (%)	3 years (*n* = 625)	11.83	23.96	76.04	1431.63	<0.001
	4 years (*n* = 4659)	88.17	51.64	48.36		
Child sex (%)	Boys (*n* = 3335)	63.11	58.18	41.82	140.42	<0.001
	Girls (*n* = 1949)	36.89	31.55	68.45		
Disability level (%)	Severe (*n* = 3839)	72.63	42.54	57.46	117.34	<0.001
	Mild (*n* = 1446)	27.37	63.83	36.17		
Disability type (%)	ASD (*n* = 421)	7.96	62.23	37.77	1641.98	<0.001
	Cerebral palsy (*n* = 1865)	35.28	47.29	52.71		
	ID (*n* = 814)	15.40	100	0		
	Language Disorder (*n* = 1315)	24.88	43.50	56.50		
	Hearing impairment (*n* = 646)	12.22	0	100.00		
	Others (*n* = 225)	4.26	11.56	88.44		
Monthly income (KRW 10,000; %)	150–299 (*n* = 1641)	27.44	64.96	35.04	2017.66	<0.001
	300–400 (*n* = 2269)	42.94	23.62	76.38		
	401–600 (*n* = 1090)	20.62	87.43	12.57		
	>601 (*n* = 285)	9.00	0	100.00		

Note. ASD = autism spectrum disorder, ID = intellectual disabilities, PAP = physical activity participation. A chi-square analysis was conducted to examine group differences in demographic variables.

**Table 2 healthcare-13-01081-t002:** Information of PA participation in PACD (*n* = 2556).

PA frequency (%)	Two times/week		More than three times/week				Almost everyday	
	33.0		43.60				22.60	
PA minutes	Mean				Median			
	34.78				30.00			
PA places (%)	Home	Outside		Private PA center	Public PA center	Disability PA center		Others
	38.99	14.26		10.80	15.58	18.54		1.83
PA types (%)	Walking	Stretching		Balance activity	Bicycling or tricycling	Swimming		Others
	14.26	25.94		32.55	15.58	6.76		4.92

Note. PA = physical activity, PACD = infants and toddlers with disabilities.

**Table 3 healthcare-13-01081-t003:** The percentages of parent-reported barriers to physical activity participation among inactive PACD (*n* = 2730).

Economic Issue	No Time	Lack of Disability Programs	Lack of Professionals Knowledgeable About Disabilities	Lack of Information	No Community-Based PA Center	PA Is Not Priority	Others
1.97	3.83	24.46	28.78	9.91	5.00	19.75	6.31

Note. PA = physical activity, PACD = infants and toddlers with disabilities.

**Table 4 healthcare-13-01081-t004:** Differences in socio-ecological factors of PA between groups.

		All (*n* = 5286; %)	PAP (*n* = 2556; %)	No PAP (*n* = 2730; %)	χ^2^	*p*-Value
Interpersonal level						
Brothers and sisters	Yes	46.06	63.48	35.46	414.19	<0.001
	No	53.94	36.52	64.54		
Grandparents	Yes	11.17	7.97	92.03	3728.21	<0.001
	No	88.83	46.56	53.44		
Organizational level						
Physical therapy	Yes	35.28	47.29	52.71	15.24	<0.001
	No	64.72	48.95	51.05		
Pre-kindergarten	Yes	37.36	56.94	43.06	100.71	<0.001
	No	62.64	43.25	56.75		
Environmental level						
Living area	Rural	51.57	52.90	47.10	17.46	<0.001
	Urban	48.43	43.53	56.47		
Government support	High	42.64	54.33	45.67	39.30	<0.001
	Low	57.36	43.91	56.09		

Note. PAP = physical activity participation. A chi-square analysis was conducted to examine group differences in socio-ecological variables.

## Data Availability

Data are available upon request.

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
