# Peer review of "The Socio-Ecological Factors of Physical Activity Participation in Preschool-Aged Children with Disabilities"

_healthcare, 2025, doi:10.3390/healthcare13091081_

Round 1

Reviewer 1 Report

Comments and Suggestions for Authors

This is a timely and valuable contribution to the field of pediatric disability research. The study tackles an important and often overlooked topic: what influences physical activity among infants and toddlers with disabilities (ITWD), using a socio-ecological perspective. This focus on the early years is especially important, as early movement experiences can shape development and long-term health. There's a clear need for more work in this area, so the insights here could be highly useful for designing interventions, informing policy, and supporting families.

Comment 1: Methodological Strengths and Suggestions
The use of a large, nationally representative dataset (Disability Status Survey) gives this study strong credibility and relevance. The decision tree approach (CHAID) is a smart choice for uncovering hierarchical relationships. Still, it would help to include a bit more detail about how the model was set up—like how the cross-validation was handled or why specific significance thresholds were chosen. Also, it’s worth briefly explaining why certain variables (e.g., parental education, employment) weren’t included, just to make the analysis feel more transparent.

Comment 2: Results Presentation
The results are clearly presented overall, but adding some key statistics (Chi-square values, p-values, effect sizes) would give readers a quicker sense of the strength and relevance of the findings. Also, the decision tree could be shown in a more visual or summarized way—it might make it easier to follow than a purely text-based explanation.

Comment 3: Theoretical and Practical Implications
The discussion ties the findings nicely to existing literature, but there’s room to go further in highlighting what this means for practice. For instance, the link between physical therapy and greater physical activity suggests a need to improve early access to such services. Similarly, the role of grandparents could inspire more inclusive family-focused programs. A few practical takeaways for clinicians, educators, or community providers would strengthen the impact of the discussion.

Comment 4: Limitations and Generalizability
There’s no clearly marked limitations section, and that would be helpful. A few things worth noting:

  • The use of caregiver-reported data, which can introduce bias.

  • Cultural context—e.g., the prominent role of grandparents in South Korea might not apply elsewhere.

  • The absence of objective PA measures (like accelerometry) means the data relies entirely on self-report.

Mentioning these briefly would give the paper a more balanced perspective.

Comment 5: Suggestions for Minor Revisions

  • Double-check that all references are correctly formatted and relevant—there seem to be a few duplicates or misplacements.

  • Be consistent in how you use and define terms like “physical activity participation” (PAP).

  • If the sample only includes 3- and 4-year-olds, consider whether the term “infants and toddlers” in the title might be misleading.

Comments on the Quality of English Language

The manuscript is generally understandable and presents the research findings in a clear and organized structure. However, the quality of English can be improved in several areas to enhance clarity, fluency, and professional tone. While the overall meaning is conveyed, there are multiple instances of grammatical errors, awkward phrasing, and non-idiomatic constructions that may hinder readability for an international audience.

Some examples include:

  • Incorrect or awkward phrasing:

    • “Grandparents were the mostly associated factor…” → should be revised to “Grandparents were the most strongly associated factor…”

    • “Lack of disability knowledgeable professionals” → should be rephrased as “lack of professionals knowledgeable about disabilities.”

  • Wordiness and redundancy: Several sentences, particularly in the introduction and discussion, could be more concise. The overuse of linking phrases like “Moreover,” “In addition,” or “Furthermore” can be streamlined to improve flow.

  • Article usage and verb agreement: There are multiple cases where articles (a, an, the) are missing or misused, and subject-verb agreement is occasionally inconsistent.

Recommendation: A thorough professional language review or copyediting is strongly advised before publication. Attention should be paid not only to grammar and syntax, but also to improving sentence structure and academic tone for better coherence and readability. Improving the English will allow the important findings of the study to be communicated more effectively and with greater impact.

Reviewer 2 Report

Comments and Suggestions for Authors

I reviewed your piece with interest and have only a few comments for your consideration. First, why did you use the 2020 survey data when the 2024 report was available? Second,  I wonder if you could raise the visibility of the fact that your research contribution arises from there having been few studies using the socio-ecological model to explore the role of various factors affecting PA among ITWD under 5? And, can you say, as I assume was the case, that you used South Korea presumably because the sample was conveniently available? You note your study's niche on lines 78-80 but making it much more salient would help your readers more readily situate the rationale and contribution of your study. Second, you say little about relative severity of disability but it would appear to matter (as you do note in passing) in children's capacity to undertake PA? Do you have any data that could provide more insight on this rather than the binary categories and forms of PA you highlight? Third, I would be interested to learn more about who completed the surveys? You note in line 94 that its authors employed a "caregiver-proxy" approach?  I also would have found it useful to learn more about the percentage of grandparents who played that role since that group figured in your findings? Fourth, enrollment (or not) in OT and PT tells your readers nothing about the character of those interventions? Did the survey provide none? Fifth, any idea why so high a percentage of participants were 4 year olds? Last, I wonder if it might be useful to lay out some potential steps that might be taken to address the barrier you identify at each analytical scale?  Right now, your conclusion on this count is broad and somewhat vague.

Comments on the Quality of English Language

The authors should be encouraged to eliminate their use of passive voice throughout this MS. And they employ it a lot! "Over" in line 195 should be "during."
